# Dendritic Cell Vaccines: A Promising Approach in the Fight against Ovarian Cancer

**DOI:** 10.3390/cancers14164037

**Published:** 2022-08-21

**Authors:** Aarushi Audhut Caro, Sofie Deschoemaeker, Lize Allonsius, An Coosemans, Damya Laoui

**Affiliations:** 1Laboratory of Myeloid Cell Immunology, VIB Center for Inflammation Research, 1050 Brussels, Belgium; 2Laboratory of Cellular and Molecular Immunology, Vrije Universiteit Brussel, 1050 Brussels, Belgium; 3Laboratory of Tumor Immunology and Immunotherapy, Department of Oncology, Leuven Cancer Institute, KU Leuven, 3000 Leuven, Belgium

**Keywords:** ovarian cancer, dendritic cell vaccines, immunotherapy, T cells, tumor-antigens

## Abstract

**Simple Summary:**

With an overall 5-year survival of only 20% for advanced-stage ovarian cancer patients, enduring and effective therapies are a highly unmet clinical need. Current standard-of-care therapies are able to improve progression-free survival; however, patients still relapse. Moreover, immunotherapy has not resulted in clear patient benefits so far. In this situation, dendritic cell vaccines can serve as a potential therapeutic addition against ovarian cancer. In the current review, we provide an overview of the different dendritic cell subsets and the roles they play in ovarian cancer. We focus on the advancements in dendritic cell vaccination against ovarian cancer and highlight the key outcomes and pitfalls associated with currently used strategies. Finally, we address future directions that could be taken to improve the dendritic cell vaccination outcomes in ovarian cancer.

**Abstract:**

Ovarian cancer (OC) is the deadliest gynecological malignancy in developed countries and is the seventh-highest cause of death in women diagnosed with cancer worldwide. Currently, several therapies are in use against OC, including debulking surgery, chemotherapy, as well as targeted therapies. Even though the current standard-of-care therapies improve survival, a vast majority of OC patients relapse. Additionally, immunotherapies have only resulted in meager patient outcomes, potentially owing to the intricate immunosuppressive nexus within the tumor microenvironment. In this scenario, dendritic cell (DC) vaccination could serve as a potential addition to the therapeutic options available against OC. In this review, we provide an overview of current therapies in OC, focusing on immunotherapies. Next, we highlight the potential of using DC vaccines in OC by underscoring the different DC subsets and their functions in OC. Finally, we provide an overview of the advances and pitfalls of current DC vaccine strategies in OC while providing future perspectives that could improve patient outcomes.

## 1. Introduction

Among a myriad of gynecological malignancies, ovarian cancer (OC) is the deadliest in developed countries [1,2]. It has the seventh-highest mortality rate among women diagnosed with cancer worldwide [3]. It is a silent killer, metastasizing throughout the peritoneal cavity before causing symptoms. Furthermore, the symptoms presented by OC patients are rather vague, including abdominal distension, abdominopelvic pain, and nausea which can be associated with other pathologies, thereby delaying timely detection [4]. Consequently, a vast majority of OC patients are only diagnosed in stages III (51%) or IV (29%), resulting in a poor overall 5-year survival rate of only 30% [5,6].

Currently, radical debulking surgery in combination with platinum-based (neo-) adjuvant chemotherapy is still the gold standard for OC treatment [1,4,5,7]. However, at advanced stages, the patients face recurrence and chemotherapy resistance. In wake of this, targeted therapies have emerged as a therapeutic advancement. Bevacizumab, an anti-vascular epithelial growth factor (anti-VEGF) monoclonal antibody, is now an integral part of treatment in both first-line therapy and relapsed patients [1,4,5,7]. Furthermore, the use of poly ADP-Ribose polymerase (PARP) inhibitors, which inhibit the action of DNA repair enzymes, has shown beneficial outcomes in OC patients with and without mutations in DNA homologous repair machinery, such as BRCA gene mutations. As such, PARP inhibitors are typically used as maintenance therapy resulting in a significant prolongation of progression-free survival (PFS) [1,4,5,7,8]. Despite all these therapeutic advances, approximately 80–85% of the advanced-stage patients still relapse [1,7], indicating the urgent need for novel therapies against OC.

## 2. Immunotherapies in OC

The interest in immunotherapy for OC treatment has been boosted, particularly since it was shown that intra-tumoral (IT) T cells, especially CD8^+^ T cells, in treatment naïve OC patients were associated with improved clinical outcomes and survival [9,10,11,12]. Zhang et al. showed that the overall 5-year survival rate was 38% in patients with IT T cells compared to 4.5% among patients without IT T cells [9]. Although these studies stress the crucial role IT T cells play in OC, their effective anti-tumor responses are often hindered by several obstacles, a prominent one being the presence of immune checkpoints.

Several immune checkpoints are expressed on OC-infiltrating T cells including PD-1, CTLA-4, TIM-3, LAG-3, BTLA, and other co-inhibitory receptors that function to modulate immunosuppressive effects [12,13]. Consequently, OC patients with high PD-L1 expression had a significantly lower 5-year survival rate (overall and progression-free) compared to patients with low PD-L1 expression [12]. A significant inverse correlation was also observed between the number of IT CD8^+^ T cells and PD-L1 expression [12]. These studies theoretically hint toward the validity of using immune checkpoint blockade (ICB) therapies in OC to restore the anti-tumoral function of IT T cells [9,10,11,14]. In contrast, ICB therapies using inhibitory antibodies against PD-1 and PD-L1 showed meager outcomes with a median response rate ranging from 6% to 22% [15,16,17]. The exact resistance mechanisms to these ICBs are still under investigation, although the pre-existence of a complex multi-faceted immunosuppressive nexus in OC may serve as a potential explanation [18]. Surprisingly, even in PD-L1 positive OC patients, anti-PD-1 treatment resulted in a poor overall response rate [17,19].

With T cells as the forefront modulators of anti-tumor immunity in OC, another promising immunotherapeutic strategy with immense potential is adoptive T-cell therapy (ACT) [20,21,22]. ACT is a personalized immunotherapy that is based on the intravenous re-administration of autologous, *ex vivo*-expanded T cells (retrieved either from the tumor tissue or peripheral blood) in combination with a high dose of Interleukin-2 (IL-2) and preceded by a non-myeloablative lymphodepleting chemotherapy regimen [18,20,21,22]. While ACT has shown stunning potency in metastatic melanoma patients, its success in OC patients is still minor [20,23]. In a pilot study involving six advanced OC patients, ACT was shown to be beneficial in targeting existing tumor lesions but failed to prevent metastases. This could be attributed to exhaustion of the tumor-infiltrating lymphocytes, poor or insufficient *ex vivo* expansion, or intra/inter-lesion heterogeneity of OC [23]. Currently, 15 phase I/II clinical trials have been registered on ClinicalTrials.gov (accessed on 15 August 2022) to scrutinize the therapeutic potential of ACT in OC patients.

Engineered T-cell receptor (TCR) and chimeric antigen receptor (CAR) T-cell therapies are two types of ACT that have also been investigated in OC. TCR T-cell therapy uses engineered antigen-specific alpha-beta chain heterodimers that can recognize specific antigens presented by HLA molecules. On the other hand, CAR T-cell therapy uses antigen-binding regions of antibodies that are fused to intracellular T-cell signaling domains, thereby allowing them to recognize surface antigens independent of HLA presentation. However, while CAR T cells can only elicit an immune response to surface antigens, TCR T cells can recognize antigens from any subcellular compartment [24]. 32 Phase I/II clinical trials of CAR T-cell therapies have been registered in ClinicalTrials.gov (accessed on 15 August 2022) directed towards several antigens including B7-H3, TAG72, ALPP, CD133, C-met, FRα, FRβ, HER2, mesothelin, and MUC-1. The first Phase I clinical trial of CAR T-cell therapy in recurrent OC (NCT03585764) was targeted toward FRα. While the therapy was safe and only resulted in grade 1 or 2 toxicities, no reduction in tumor burden was observed in any of the six tested patients [25]. This and several other clinical trials in OC targeting different antigens have demonstrated the poor success of CAR T-cell therapies in OC, much like in several other solid tumors [24,26]. TCR T-cell therapy in OC on the other hand is still in its early phase with 19 Phase I/II clinical trials registered in ClinicalTrials.gov (accessed on 15 August 2022).

From the limited benefits of ICB and ACT, it appears that the keys to the success of immunotherapy in OC include weakening the immunosuppressive immune compartment, boosting the effector T-cell responses, and stimulating the antigen-presenting cells (APCs) [22]. Correll et al. demonstrated that dendritic cell (DC) vaccination could reduce the regulatory T-cell (Treg) compartment in the peripheral blood of malignant melanoma patients, which in turn allowed T-cell effector functions to be restored [27]. Hence, using DCs as a vaccination agent for cancer could potentially achieve all three keys to successful immunotherapy.

DCs are popularly known as the immune system’s professional APCs and are paramount for initiating anti-tumor immune responses [28]. Harvesting the potential of DCs as a vaccine in cancer to prime T cells against tumor antigens is a long-standing concept. The first pilot study for DC vaccination was performed in 1996 in four follicular B cell lymphoma patients using autologous antigen-pulsed DCs. All patients showed positive outcomes with measurable anti-tumor immune responses, including one complete and one partial tumor regression [29]. Hence, the immense potential DC vaccines hold as an immunotherapeutic agent in cancer is no surprise and has been investigated in multiple cancers. To date, more than 400 clinical trials of DC vaccines have been registered on ClinicalTrials.gov in different cancers (accessed on 15 August 2022). Results of completed studies highlight that DC vaccines are relatively safe and result in positive patient outcomes, although in only a minority of patients [30,31,32,33]. Consequently, in 2010, the FDA approved the first DC vaccine (Sipuleucel-T) against metastatic prostate cancer [30,34]. These studies have served as an inspiration to also investigate DC vaccines in OC. In this review, we aim to evaluate the use of DC vaccines in OC while also highlighting studies from other cancers that could provide insights into bettering their use in OC.

## 3. DC Subsets and Functions in Inflammation and Cancer

In 1973 Ralph Steinman and Zanvil Cohn observed the presence of adherent nucleated cells with distinct morphological features within murine spleens and lymph nodes (LN). A characteristic feature of these cells was the peculiar arrangement of their cytoplasm into pseudopods of varying lengths, widths, numbers, and forms, which made them annotate these cells as the “dendritic cells” (DCs) [35].

DCs are the most efficient APCs and serve as the bridge between the innate and adaptive immune systems by capturing, processing, and presenting antigens to T cells [36,37]. They are the dominant cell type in eliciting antigen-specific immunity and tolerance, despite their sparse numbers in the body [37]. DCs can be subdivided into three main subsets; the conventional or classical DCs (cDCs, also referred to as myeloid DCs), monocyte-derived DCs (moDCs), and plasmacytoid DCs (pDCs) [37,38,39,40]. cDCs can further be sub-divided into cDC1, cDC2 and migratory DCs (migDCs) [37]. An overview of the panel of markers used to identify the different DC subsets in mice and humans as well as their key functions is listed in Figure 1.

### 3.1. cDC1

It is well established that cDC1s are proficient in cross-presenting exogenous antigens via MHC-I to CD8^+^ T cells [37,41,42,43]. This is particularly important in cancer immunity, as the rejection of murine B16-OVA (melanoma) and 1569 fibrosarcoma was lost when MHC-I expression on cDC1s was absent [44]. Moreover, in mouse and human cancers, cDC1s have been appreciated for their potent ability in regulating cancer immunotherapy responses by activating tumor (neo)antigen-specific CD8^+^ T cells [45]. Furthermore, responses to ICBs (including anti-TNFRSF9 (also known as 4-1BB) [46], anti-PD-1 [46,47], or anti-CTLA4 [47]), in mice are dependent on cDC1s. CD103^+^ cDC1s, although representing a minor tumor-infiltrating population in murine melanoma, were shown to be the predominant vehicles for transporting tumor antigens (TAs) to the tumor-draining lymph nodes (tdLNs) and were key in promoting anti-PD-L1 mediated anti-tumor immunity [48]. That said, cDC1s are also potent producers of Interleukin-12 (IL-12) and can induce NK and CD8^+^ T-cell cytotoxicity and Interferon-gamma (IFNγ) production [49] in addition to driving T helper type 1 (Th1) CD4^+^ T-cell responses [43]. cDC1s are also key in the early priming of CD4^+^ T cells via MHC-II against tumor-derived antigens [44]. Eickhoff et al. have demonstrated that cDC1s are critical for the delivery of CD4^+^ T cell help to CD8^+^ T cells and their absence leads to poor differentiation of memory CD8^+^ T cells in viral infections [50]. As a positive feedback loop, IFNγ augments cDC1-mediated IL-12 production, strengthening antigen cross-presentation [49].

### 3.2. cDC2

Classically, cDC2s are known to prominently activate CD4^+^ helper T cells, particularly T helper type 2 (Th2) [51,52] and T helper 17 (Th17) cells [53,54,55]. They are ontogenically distinct from cDC1s and cannot functionally compensate for cDC1 deficiencies [38]. LLC-OVA lung cancer-bearing mice vaccinated prophylactically with tumor-derived cDC2s showed a significant reduction in tumor growth and weight compared to non-vaccinated mice, particularly by cDC2-mediated polarization of the Th17 immunity [53]. Similarly, Neubert et al. showed that antigen targeting to cDC2s in the presence of anti-CD40 and Toll-like receptor 3 (TLR3) ligand resulted in the initiation of protective immunity against B16F10-OVA melanoma in both the prophylactic and therapeutic setting [56]. Moreover, cDC2s were shown to further activate existing CD8^+^ T cells upon anti-CD40 therapy [57].

However, three major bottlenecks exist in understanding cDC2 functions. First, no unique cDC2-specific marker has been identified to date, hence hampering the understanding of the *in vivo* contribution of cDC2s to tissue immune responses using conditional depletion models [58]. Second, there is an overlap in the currently used cDC2 markers and phenotypic characteristics with other myeloid compartments such as moDCs and macrophages, thereby complicating the strict contribution of cDC2s rather than other myeloid cells to functional inferences [59]. Third, the cDC2 compartment is rather heterogenous harboring multiple sub-populations [37,39,60], which could mean that each subset may be attributed to a different functionality. Indeed, several cDC2 subsets have been described in the literature to date in mice and/or humans either in steady state or inflammation, including CD301b^+^ and CD301b^−^ cDC2s [61,62], Notch2-dependent CX3CR1^lo^Esam^hi^ cDC2 [63], CLEC12A^−^Esam^+^Tbet^+^ cDC2A and CLEC12A^+^Esam^−^Tbet^−^ cDC2B [64], IRF8^+^CD64^+^ inflammatory cDC2s [65], AXL^+^SIGLEC6^+^ DCs (AS-DCs) in humans (or transitional DCs (tDCs) in mice) [66,67], and Klf4-dependent IRF4^+^ cDC2s [68].

While these different cDC2 sub-populations have been described in different immune settings and annotated accordingly, their phenotypes may overlap to some extent and further concurrent and comprehensive investigation is necessary to gain clarity. Moreover, this could also be relevant in terms of DC vaccines wherein the use of only the most potent cDC2 subpopulation may result in better patient outcomes rather than the whole cDC2 compartment which may harbor some anti-inflammatory sub-populations.

### 3.3. migDC

Migratory DCs (migDCs, also termed DC3 [37,69], mregDC [70,71], LAMP3^+^ DCs [72]) represent a distinct mature cell state that can arise in both cDC1s and cDC2s upon antigen sensing or uptake [70,71]. Typically, migDCs are characterized in mice and humans by the expression of CCR7 [58,73] which is the chemotactic driver that can guide them towards the LNs [58,71] and LAMP3 (only in humans) [72]. MigDCs are non-lymphoid tissue DCs that can migrate to the tdLN via the lymphatic system rather than the blood. In an inflammatory setting, migDCs loaded with antigen migrate to the T-cell zones of the LNs wherein they can prime CD4^+^ and CD8^+^ T cells. This migratory phenotype is also characterized by the upregulation of MHC-II, costimulatory molecules, and secretion of inflammatory cytokines, which further drive the inflammatory T-cell responses [58,74]. Besides directly priming T cells in the LNs, migDCs can also transfer their antigens to the LN resident DC, which can prime T cells [74].

### 3.4. pDC

pDCs are primarily involved in mounting anti-viral responses [75]. They are less adept than cDCs in priming T cells but are potent producers of type-I Interferons (IFNs), which boost anti-tumor immunity [37]. Interferon-alpha (IFNα) produced by the pDCs serves as an inhibitor of cancer cell proliferation, metastasis, and angiogenesis. Moreover, pDCs induce direct cytotoxicity of tumor cells through the expression of TNF-related apoptosis-inducing ligand (TRAIL) and Granzyme B. Furthermore, pDCs can produce CCR5 which mediates the recruitment of NK cells to the TME. In head and neck squamous cell carcinoma, OX40^+^ pDCs in the TME showed enhanced immunostimulatory and cytolytic activities that synergized with cDCs in eliciting potent TA-specific CD8^+^ T-cell responses [76]. Similarly, in melanoma patients, pDCs attract cytolytic lymphocytes including CD8^+^ T cells, CD56^+^ T cells, and γδ T cells, more potently compared to cDC2s [77]. Contrastingly, pDCs under the influence of primary breast tumor cell-derived granulocyte-macrophage colony-stimulating factor (GM-CSF) were shown to present a tolerogenic phenotype and aid in tumor progression [78].

### 3.5. moDCs

MoDCs are a controversial cell subset that is developmentally different from cDCs but manifests the properties of DCs and monocytes [37,53,79]. As ontogenically these cells are not DCs, a more apt nomenclature for them would be “monocyte-derived cells”. However, many clinical trials involving DC vaccination undertaken to date use these cells and address them as “monocyte-derived dendritic cells (moDCs)”. For the simplicity of correlating the information in this review with existing literature, we will continue to use the term “moDCs” to refer to these cells.

MoDCs expand during inflammation and can regulate T-cell responses. However, currently, the major setback in studying moDCs is their extensive overlap with cDC2s, thereby blurring their relevance in cancer immunology [37,49]. MoDCs are typically used for DC vaccinations, wherein they are generated *in vitro* using peripheral blood monocytes in the presence of GM-CSF and Interleukin-4 (IL-4). However, in contrast to *in vivo* moDCs, the *in vitro* generated moDCs encompass a heterogeneous population consisting of DC-like and macrophage-like cells [39,80]. Moreover, the resulting DC-like cells differ from cDC1s and cDC2s [39]. Pre-clinical studies have shown that MoDCs are critical in driving immune responses during chemotherapy and cell therapies [49,81]. Ma et al. have shown that moDCs are key in initiating anti-tumor immune responses after anthracycline chemotherapy by optimally presenting TAs [81].

Overall, although cDC1s, cDC2s, pDCs, migDCs, and moDCs are the major DC sub-populations in steady-state and inflammation, other DC states may exist depending on their anatomical location and exposure to external stimuli [37].

## 4. Role of DCs in OC

In OC patients, a reduced cDC1 number coincided with increased detection of the OC tumor marker CA125 [82]. In addition, higher expression of CLEC9A (a marker specific for cDC1s) in the OC tissue correlated with better overall survival (OS) [82]. In three independent cohorts of OC patients without neoadjuvant chemotherapy, it was shown that tumor-infiltrating mature migDCs (LAMP3-DCs) served as a favorable prognostic marker and were key in eliciting anti-tumor CD8^+^ T-cell responses [83].

Scarlett et al. describe the predominant leukocyte subset in advanced murine (p53/K-ras mice intra-bursally injected with Cre-recombinase), and human OC tissue to be putative DCs (CD11c^+^DEC205^+^MHCII^+^). These DCs exhibited both pro- and anti-tumoral properties, depending on the disease stage. In the early tumor stages, the DCs were immunostimulatory and drove T cell-mediated tumor control, while at advanced stages they turned immunosuppressive. Furthermore, the latter showed a significantly lower expression of MHC-II and CD40 and increased expression of PD-L1 and arginase [84]. That said, it is important to note that the markers used for identifying the said DCs overlap with macrophages. Hence, strictly only DCs being the predominant leukocyte population in OC tissue is surprising.

*In vitro* generated bone marrow-derived DCs, when cultured with VEGF-A-producing ID8 mouse ovarian tumor cells (ID8-Vegf), obtained endothelial cell features [85]. Similarly, in ID8-Vegf *Defb29* (ID8 cells producing high amounts of VEGF-A and beta-defensins) tumors and ascites, MHC-II^+^CD11c^+^DEC205^+^CD8a^+^ DCs with low expression of co-stimulatory molecules were found to be increasingly attracted to the tumor via CCR6 and enhanced tumor growth by achieving endothelial cell-like features and aiding in tumor vasculogenesis [85].

When the whole DC compartment was ablated using CD11c-DTR mice immediately followed by subcutaneous inoculation of ID8-Vegf *Defb29* cells, a 3-fold decrease in tumor size was observed after two months [86]. Additionally, with conditional depletion of DCs using CD11c-DTR mice as the tumors progressed from early to advanced stages (immunological control to tumor escape), tumor progression was observed to be slower [84]. However, it is noteworthy to point out that CD11c is also expressed on some macrophages which will also be depleted in these studies and could contribute to the observed clinical outcome [59].

Accumulation of pDCs in ovarian tumors has been associated with early relapse and poor patient prognosis [87]. They can induce Interleukin-10 (IL-10) production by T cells which can result in poor cDC-mediated T-cell activation. Additionally, pDCs from malignant ascites of OC patients were shown to advance tumor angiogenesis via the production of Tumor necrosis factor-alpha (TNFα) and Interleukin-8 (IL-8) [88]. Ascites-derived pDCs from OC patients were also shown to induce CD8^+^ Treg responses in an IL-10-dependent manner which could in turn diminish TA-specific effector T-cell functions [89]. Tumor pDCs could also induce immunosuppression by inducing expansion of ICOS^+^FoxP3^+^ Tregs via ICOS-L co-stimulation [90].

MoDCs generated from patient ascites-derived CD14^+^ cells and differentiated *in vitro* using GM-CSF and IL-4 were shown to mitigate angiogenesis *in vivo* in NOD-SCID mice in an IL-12-dependent manner [88]. Additionally, MoDCs could efficiently induce CD8^+^ T cells by antigen cross-presentation and co-stimulation through IL12p70. They also induced cytotoxic activity in CD8^+^ T cells by the production of Granzyme A, Perforin, and IFNγ [91]. Segura et al. described inflammatory DCs (which share a gene signature with moDCs) in ovarian ascites to induce differentiation of naïve CD4^+^ T cells and Th17 polarization [92].

These studies together represent the diverse functions DCs can undertake in OC, either attributed to the different stages of tumor progression or the DC subset itself. Hence, further research on DC diversity, especially owing to the high heterogeneity of cDC2s, is necessary to streamline their roles in OC immunity. This in turn could aid in the development of robust DC-based therapies against OC.

## 5. DCs Vaccines in OC

Adjuvant DC vaccines have shown prolonged therapeutic effects resulting in improved survival, or reduced chances of relapse in melanoma [93], glioblastoma [94], prostate cancer [34], and renal cell carcinoma [94], although in only a small subset of patients. Yet, DC vaccination is relatively safe with minimal toxicities, often less than chemotherapies and ICBs [94].

The typical steps involved in manufacturing DC vaccines include (1) retrieval of immature cells or mature DCs from the patient; (2) *in vitro* differentiation and maturation of immature cells into DCs using a maturation cocktail; (3) antigen-loading of the mature DCs; (4) administration into patients via different routes [95] (Figure 2). Currently, 24 DC vaccination clinical trials in OC have been registered on ClinicalTrials.gov (accessed on 15 August 2022) (Table 1). The published outcomes of different clinical trials and pilot studies undertaken in OC so far (accessed on 15 August 2022) have been summarized in Table 2.

From the 23 clinical trials and pilot studies listed in Table 2, three Phase II trials addressing if the addition of moDC vaccination to the standard-of-care (SOC) treatment provides added benefit in comparison to the SOC alone have been performed by the group of Prof. Spisek. In the first phase II clinical trial (NCT02107950) in stage III-IV platinum-sensitive OC patients relapsing after first-line chemotherapy, the added benefit of moDC vaccination in addition to SOC was investigated with moDC vaccination being initiated after two cycles of chemotherapy [94]. Although the moDC vaccination in combination with chemotherapy did not improve PFS, the OS was prolonged by 13.4 months [94]. In the second phase II clinical trial (NCT02107937), stage III OC patients were either injected with moDC vaccine in parallel with chemotherapy (Arm 1), moDC vaccine sequentially after chemotherapy (Arm 2), or chemotherapy alone (Arm 3). Patients in Arm 2 who received moDC vaccination after chemotherapy treatment had a significantly prolonged PFS and a 60% reduced risk of death compared to patients in Arm 3 only receiving chemotherapy treatment. This indicates that moDC vaccination can lead to additional benefits when added to SOC. When comparing PFS between the chemotherapy alone group (Arm 3) and moDC vaccination administered in parallel with chemotherapy (Arm 1), there was, however, no difference, indicating that the moDC vaccine functions better after the tumor burden has been reduced using chemotherapy [96]. A retrospective analysis of the same clinical trial reported that OC tumors with high CD8^+^ T-cell infiltration (“hot tumors”) with a high tumor mutational burden (TMB), profit more from chemotherapy, while less infiltrated or “cold” tumors with a low TMB rely on moDC vaccination (irrespective of when moDC vaccination was given) to kick-start anti-cancer responses [97]. In another study further evaluating the potential prognostic markers for moDC vaccine response, the presence of Tregs in the blood was found to be a negative prognostic factor in OC patients. Indeed, only in patients with a low number of Tregs did the CD8^+^ T cells increase after moDC vaccination [98]. In addition, patients that had received moDC vaccination (both Arms 1 and 2) also showed the presence of antigen-specific cytotoxic T cells in the blood [97].

Additionally, in another randomized phase II clinical trial (NCT01068509), the outcome of 56 OC patients in response to SOC treatment versus mannan-mucin 1 fusion protein-loaded moDC vaccine as maintenance therapy was assessed. OC patients in the first (CR1) or second clinical remission (CR2) were included in this trial and randomized 1:1. Overall no improved PFS or OS was observed between patients receiving SOC treatment and moDC vaccine. However, a comparative analysis of clinical outcomes of CR1 and CR2 patients showed that CR2 patients receiving the DC vaccine compared to SOC therapies had improved PFS (13 months versus 5 months, respectively) and median OS (not reached at 42 months versus 26 months, respectively) [99]. This study highlights that stratification of patients into different cohorts during the trials could give an insight into the likely-to-respond patients which could in turn aid in targeting the right patient group for optimal therapy responses.

Overall, these studies provide evidence for the added benefit of moDC vaccination, but also show that improvement is still possible. This could be either at the level of the type of DC vaccine/manufacturing procedure, the combination of therapies used along with the DC vaccine, or markers used to identify responsive or responding patients.

### 5.1. Influence of the Type of Antigen/Antigen Loading in DC Vaccination on the Immunological Outcome of Patients

As presented in Figure 2, moDC vaccines need to be loaded with antigens to be able to mount an anti-tumor immune response. The Phase II clinical trials NCT02107950 and NCT02107937 described above have made use of monocytes from PBMCs that were cultured in a medium with GM-CSF and IL-4, followed by pulsing with hydrostatic pressurized human OV-90 and SKOV-3 cell lysate and maturation with poly(I:C) (TLR3 ligand) [96,97]. Therefore, these trials are unique in using allogeneic cancer cell lines as a source of tumor antigens.

A second method commonly applied is the use of autologous tumor as a source for tumor antigens. Indeed, in a phase I clinical trial (NCT01132014) hypochlorous acid (HOCl) treatment followed by freeze-thaw cycles was used to prepare the autologous tumor cell lysate. Autologous *in vitro* differentiated moDCs were then pulsed with the tumor cell lysate in the presence of lipopolysaccharide (LPS) and IFNγ [100]. A significant enhancement of TA-specific T cells was observed post-vaccination in the peripheral blood at end of the study in four out of five patients whereas only one out of five patients showed clinical benefit. Furthermore, at the end of the study, the moDCs were found to strongly induce Th1 responses capable of secreting high levels of IFNγ. Importantly, the IFNγ producing CD3^+^ T cell to FoxP3^+^ Treg ratio in the peripheral blood increased 1.5–3.6 fold in four out of five patients. A 0.5-fold increase in serum IL-2, IL-1β, IL-15, TNFα, CCL2, CCL3, CXCL9, and eotaxin were observed while the levels of serum IL-10 dropped post-vaccination. Interestingly, the HOCl-oxidized tumor lysate-pulsed moDC vaccine was able to cross-present TAs via MHC-I to CD8^+^ T cells efficiently [100].

Additionally, in a phase I/II clinical trial (KCT0000831) performed by Baek et al., moDCs were pulsed with autologous whole tumor lysate and keyhole limpet hemocyanin (KLH) and combined with IL-2 treatment in 10 OC patients. The KLH is a foreign helper protein known to enhance anti-tumor immunity. Increased cytotoxic NK cell (CD16^+^CD56^+/^^−^) abundance, as well as NK cell activity, was observed in half of the vaccinated patients [101].

Alternatively, moDCs can also be pulsed with specific neoantigen peptides or proteins. For example, Morisaki et al. reported that a stage III chemo-resistant and recurrent OC patient with malignant ascites benefited from a neoantigen peptide-loaded moDC vaccine. After four rounds of vaccination, CA125 levels declined and tumor-associated symptoms such as respiratory discomfort were alleviated. This was accompanied by the presence of neoantigen-specific CD8^+^ T cells in the ascites of the patient, though the patient died 15 months after the start of the moDC vaccination regimen [102].

Additionally, Block et al. reported an early phase I clinical trial (NCT02111941) using moDCs pulsed with OC antigen FRα and treated with a p38 MAPK inhibitor in combination with IL-15 to drive a Th17 anti-tumor response [103]. Prior *ex vivo* studies from the group demonstrated that this moDC vaccine production protocol resulted in reduced PD-L1 expression and indoleamine 2,3-dioxygenase (IDO) function in DCs, which in turn favored Treg reduction and elicited OC antigen-specific Th1/Th17 responses [104]. Moreover, the moDCs upregulated the expression of CCR7, indicating their LN migratory potential. The moDCs also consistently expressed high levels of CCL22 which is a chemo-attractant for CCR4^+^ T cells [104]. In the clinical trial, similar effects were observed in the vaccinated patients wherein Th1, Th17, and anti-FRα antibody-dependent cell-mediated cytotoxicity (ADCC) were associated with prolonged recurrence-free survival (RFS) [103]. Other peptides that have been used for pulsing of moDCs for ovarian cancer are, for example, p53 peptide [105], Wilm’s Tumor 1 (WT1) peptide [106], a combination of Her-2/neu and mucin 1 [107] or a combination of hTERT, Her-2/neu, and PADRE peptides [108].

On the other hand, the phase I study performed by Loveland et al. [109] and the phase II study performed by Gray et al. (NCT01068509) [99] use moDCs pulsed with oxidized mannan coupled with mucin 1-glutathione S-transferase fusion protein. Hernando et al. have used electroporation of FRα mRNA to introduce the antigen in the moDCs [110] and Coosemans et al. have electroporated moDCs with WT1 mRNA [111].

Overall, the above-mentioned clinical trials highlight that the antigen and antigen loading methods used in DC vaccine manufacturing can influence the generation of vaccine-mediated antitumor immunity including TA-specific T-cell responses, though comparative studies are missing to understand the best methodology to be used.

### 5.2. Influence of Combination Therapies with DC Vaccine on the Immunological Outcome of Patients

While the phase II studies described above already indicated the added benefit of moDC vaccination to the SOC chemotherapy, the efficacy and immunological outcome of other combinations were also evaluated.

Kandalaft et al. reported two pilot clinical studies wherein moDC vaccination was tested in advanced OC patients in combination with bevacizumab and cyclophosphamide [112]. Four out of six patients experienced clinical benefits. Moreover, Treg abundance in circulation was lowered, TA-specific T cells were increased, and IgG and IgM seropositivity improved [112].

In the phase I clinical trial (NCT01132014), the effectiveness of HOCl-oxidized lysate-pulsed moDC vaccination (Arm 1) in comparison with moDC vaccine in combination with either bevacizumab (Arm 2) or bevacizumab and cyclophosphamide (Arm 3) was assessed [113], wherein CD8^+^ and CD4^+^ T cells were found to be increased in the peripheral blood in all arms post-vaccination. Moreover, neoepitope-specific CD8^+^ T-cell responses were detected in vaccinated patients, wherein the T cells produced IFNγ, TNFα, and IL-2. Interestingly, peripheral blood T cells from vaccinated patients were able to more efficiently recognize autologous whole tumor cells, as well as TAs, presented on DCs in comparison to peripheral blood T cells collected before vaccination. In addition, patients exhibiting such responses presented with a significantly increased PFS. These results indicate that the immunological outcomes, in terms of T-cell responses, were induced by the moDC vaccine and are relevant for patient outcomes. Furthermore, T-cell responses in patients treated with the triple combo (Arm 3) were significantly higher than those observed in patients receiving only moDC vaccination and bevacizumab (Arm 2). In line with this, patients from Arm 3 had an improved OS compared to Arm 2. Additionally, the OS of Arm 3 was significantly higher compared to a historical control group receiving only bevacizumab/cyclophosphamide (without moDC vaccine) [113]. These results suggest the added benefit of the triple combination of cyclophosphamide, bevacizumab, and moDC vaccination above the double combination of moDC vaccination and bevacizumab.

To further improve treatment response, in the same study (NCT01132014), the combination of HOCl-oxidised autologous whole-tumor lysate-pulsed moDC vaccine, bevacizumab/cyclophosphamide, acetylsalicylic acid (ASA,) and IL-2 was assessed. Arm 4 received moDC vaccine/bevacizumab/cyclophosphamide/ASA while Arm 5 received moDC vaccine/bevacizumab/cyclophosphamide/ASA/IL-2 [114]. ASA is an inhibitor of prostaglandin E2 synthesis. Tumor-derived VEGF and prostaglandin E2 are known to elicit tumor endothelium-dependent killing of CD8^+^ T cells by expressing Fas ligand (FasL). On the other hand, IL-2 is the predominant T cell growth factor. Hence, the combination of ASA and IL-2 with a previously designed combination of moDC vaccine with bevacizumab/cyclophosphamide was assessed [114]. It was found that the patients in Arm 5 had an increased abundance of polyfunctional tumor neoantigen-specific CD8^+^ T cells capable of producing IFNγ, TNFα, perforin, and Granzyme B compared to patients in Arm 4. This was also accompanied by an increase in survival in patients from Arm 5 with 80% of the patients surviving longer than 3 years in comparison to patients in Arms 3 and 4, with only 40% of the patients surviving longer than 3 years [114].

**Table 2 cancers-14-04037-t002:** An overview of different pilot studies and clinical trials of DC vaccination with published results carried out in OC (accessed on 15 August 2022).

Phase of Study	No. of Patients	DC Generation	No. of Doses/No. of DCs per Dose	Injection Site	Combination	Response Rate	Refs.
Pilot	3	Monocytes from PBMCs were cultured in a medium with IL-4, GM-CSF, and TNFα followed by pulsing with Her-2/neu and mucin 1 peptides.	3–9 doses of 2.8–8.7 million DCs	SC near inguinal LNs	-	SD	[107]
Phase I	6	Monocytes from PBMCs were cultured in a medium with GM-CSF and IL-4; pulsed with Keyhole Limpet Hemocyanin (KLH) and autologous tumor cell lysate in the presence of GM-CSF and TNFα.	3–23 doses of 1–90 million DCs	IC near axillary LNs	-	1/6-dead from disease 3/6-PD 2/6-SD	[115]
Phase I	1	Monocytes from PBMCs were cultured in a medium with GM-CSF and IL-4 followed by pulsing with the mannan-mucin 1 fusion protein.	Multiple doses of 40 million DCs	ID and SC	-	SD	[109]
Pilot	4	Monocytes from PBMCs were cultured in a medium with GM-CSF, IL4 and TNFα followed by pulsing with tumor cell lysate and treatment with 50% polyethylene glycol.	6 doses of 10–26 million	SC near the neck or groin	IL12	1/4 patients-PD with transient reduction of CA125	[116]
Pilot	1	Monocytes from PBMCs were cultured in a medium with GM-CSF and IL-4, followed by electroporation with FRα mRNA and culturing in IL-1β, IL-6, TNFα, and prostaglandin E2.	10 doses of 2–21 million DCs	IN in the inguinal LNs	-	PR	[110]
Phase I	4	PBMCs were cultured with ionomycin and 10 µg/mL or 40 µg/mL BA7072 antigen (fusion protein containing sequences from intracellular and extracellular domains of Her-2 linked to GM-CSF.	3 doses of 10^9^ cells	IV	-	2/4-SD 2/4-PD	[117]
Phase I/II	11	Monocytes from PBMCs were cultured in a medium with IL-13 and GM-CSF, followed by maturation with membrane components of *Klebsiella pneumoniae* and IFNγ; pulsing with hTERT, Her-2/neu, and PADRE peptides.	4 doses of 35 million DCs	ID into the medial thigh	**Arm1:** DC vaccine **Arm2:** DC Vaccine + Cy *Eligible patients were also given IM Prevnar heptavalent Pneumococcal vaccine with the first dose of DCs*	**Arm1**-2/5 PD, 2/5 CR **Arm2**-2/6 PD, 4/6 CR	[108]
Phase II	21	Monocytes from PBMCs were cultured in a medium with IL-4 and GM-CSF, followed by maturation using CD40L and pulsing with p53 peptide.	4 doses of 20 million DCs	IV	**Arm1:** p53 peptide + Montanide ISA-51 + GM-CSF + IL-2 **Arm2:** DC vaccine + IL-2	**Arm1**-2/13 CR, 11/13 PD **Arm2**-2/7 CR, 5/7 PD	[105]
Pilot	6	Monocytes from PBMCs were cultured in a medium with IL-4 and GM-CSF.	3 doses of 5–10 million DCs	ID	**Arm1:** Bev/Cy + DC vaccine. **Arm2:** Bev/Cy + *in vitro* generated 5 × 10^9^ T cells	**Arm1**-2/6 SD, 2/6 PR, 2/6 PD **Arm2**-1/3 CR, 1/3 SD, 1/3 PD	[112]
Early phase I	5	Monocytes from PBMCs were cultured in a medium with GM-CSF and IL-4, followed by pulsing with hypochlorous acid (HOCl)-oxidized whole tumor lysate and maturation with LPS and IFNγ.	5 doses of 5–10 million DCs	IN in the inguinal LN	-	2/5-PD 1/5-mixed response 2/5-Improved PFS	[100]
Pilot study	2	Monocytes from PBMCs were cultured in a medium with GM-CSF and IL-4, followed by electroporation with Wilm’s Tumor 1 (WT1) mRNA and culturing with TNFα and IL-1β.	4 doses of 21 million DCs	ID in the groin	-	Improved OS after chemotherapy following cessation of DC vaccination	[111]
Phase II	26	Monocytes from PBMCs were cultured in a medium with GM-CSF and IL-4 followed by incubating with mannosylated mucin 1 protein.	3–7 doses of 25–40 million DCs	ID in the upper arm and thighs	-	2/26-PD 2/26-PR 1/26-CR	[118]
Retrospective study	56	PBMCs were cultured in a medium with GM-CSF and IL-4 followed by stimulation with OK-432 (streptococcal immunological adjuvant) and prostaglandin E2, and pulsed with either WT1, mucin 1, or CA125 proteins.	5–7 doses of 10 million DCs	ID near axial or inguinal LNs	OK-432 in patients without allergies to penicillin or other drugs	42-PD 7-SD 1-PR	[119]
Phase II	7	Monocytes from PBMCs were cultured in a medium with GM-CSF and IL-4, followed by pulsing with whole tumor lysate and stimulation with Poly I:C.	6 doses of >1 million DCs	IV	-	4-PD 2-SD 1-PR	[120]
Phase I/II	10	Monocytes from PBMCs were cultured in a medium with GM-CSF, and IL-4, and matured in TNFα, followed by pulsing with tumor lysate and Keyhole Limpet Hemocyanin (KLH).	2 doses of 40 million DCs	SC near axillary LN	DC vaccine + IL-2	5/10-CR 4/10-PD 1/10-dead from disease	[101]
Phase II	56	Monocytes from PBMCs were cultured in a medium with GM-CSF and IL-4, followed by pulsing with oxidized mannan coupled with mucin 1-glutathione S-transferase fusion protein.	6–10 doses of 60 million DCs	ID in upper arms and thighs	**Arm1:** Standard-of-care treatment **Arm2:** DC vaccine	No improved PFS or OS between Arm1 and Arm2	[99]
Phase I	25	Monocyte from PBMCs were cultured in a medium with GM-CSF and IL-4, followed by pulsing with HOCl-oxidized whole tumor lysate and maturation with LPS and IFNγ.	5 doses of 5–10 million DCs	IN in the inguinal LN	**Arm1:** DC vaccine **Arm2:** DC vaccine + Bev **Arm3:** DC vaccine + Bev/Cy	**Arm1:** 2/5 PD, 3/5 SD **Arm2:** 5/10 PD, 5/10 SD **Arm3:** 2/10 PD, 8/10 SD	[113]
Phase I/II	3	Monocytes from PBMCs were cultured in a medium with GM-CSF and IL-4, followed by maturation in TNFα and pulsing with WT1 peptide.	5 doses of 10–20 million DCs	ID into bilateral axillary parts	OK-432 lyophilized mixture of group A *Streptococcus pyrogenes*.	1/3-SD 2/3-PD	[106]
Pilot study	1	Monocytes from PBMCs were cultured in a medium with GM-CSF and IL-4 followed by maturation with TNFα and IFNα; pulsed with autologous HLA type-1 restricted neoantigen peptides.	4 doses of 5–12 million DCs	IN in the inguinal LNs	-	Improved symptoms	[102]
Early phase I	19	Monocytes from PBMCs were cultured in a medium with IL-4, GM-CSF, IL-15, and methylsulanylimidazole (p38 MAPK inhibitor), followed by maturation in TNFα, IL-1β, and prostaglandin E2 and then pulsed with 4 folate receptor-α (FRα) peptides.	5 doses of 10–20 million DCs	ID into two ipsilateral areas of the body	-	39% RFS of over 48 months from the time of enrollment into the study	[103]
Phase II	71	Monocytes from PBMCs were cultured in a medium with GM-CSF and IL-4, followed by pulsing with hydrostatic pressurized human OV-90 and SKOV-3 cell lysate; and maturation with poly(I:C) (TLR3 ligand).	10 doses of 10 million DCs	SC near the inguinal LNs	**Arm1:** Chemotherapy **Arm2:** DC vaccine + chemotherapy	Improved OS of Arm2 over Arm1	[121]
Phase I	30	Monocyte from PBMCs were cultured in a medium with GM-CSF and IL-4, followed by pulsing with hypochlorous acid (HOCl)-oxidized whole tumor cell lysate and maturation with LPS and IFNγ.	5 doses of 5–10 million DCs	IN in the inguinal LN	**Arm1:** DC vaccine + Bev/Cy **Arm2:** DC vaccine + Bev/Cy + ASA **Arm3:** DC vaccine + Bev/Cy + ASA + IL-2	**Arm1:** 8/10 >3 yr OS **Arm2:** 4/10 >3 yr OS **Arm3:** 4/10 >3 yr OS	[114]
Phase II	136	Monocytes from PBMCs were cultured in a medium with GM-CSF and IL-4, followed by pulsing with hydrostatic pressurized human OV-90 and SKOV-3 cell lysate; and maturation with poly(I:C) (TLR3 ligand).	10–15 doses of 10 million DCs	-	**Arm1:** DC vaccine in parallel with chemotherapy **Arm2:** DC vaccine after chemotherapy **Arm3:** Chemotherapy only	**Arm1:** 20.3 months PFS **Arm2:** PFS not reached by the end of the study **Arm3:** 21.4 months PFS	[96,97]

Abbreviations: PBMC, peripheral blood mononuclear cells; hTERT, human telomerase reverse transcriptase; PADRE, pan-HLA-DR binding epitope; SC, subcutaneous; IC, intracutaneous; IN, intranodal; ID, intradermal; IV, intravenous; IM, intramuscular; Bev, bevacizumab; Cy, cyclophosphamide; ASA, acetylsalicylic acid; SD, stable disease; PR, partial response; PD, progressive disease; PFS, progression-free survival; OS, overall survival; CR, complete response; RFS, recurrence-free survival.

In summary, all DC vaccination clinical trials/pilot studies carried out in OC so far used moDCs, either pulsed with specific (neo-)antigens or whole tumor lysates or electroporated with TA mRNA. Nearly all trials resulted in improved TA-specific CD8^+^ T-cell activity with enhanced IFNγ production post-vaccination, although assessment of other markers such as the presence of Tregs, NK cells, or a Th17 response could be interesting depending on the type of vaccine used. Moreover, grade >3 toxicities were rare. Despite these outcomes and the increased efficacy of moDC combination strategies, the overall success of DC vaccines in OC so far has been poor.

Interestingly, in a phase I/II clinical trial undertaken by Baek et al. with 10 OC patients, it was observed that clinical benefit was experienced mostly in patients with stable disease at the time of inclusion, while those with progressive disease showed no clear clinical benefits [99,101].

Overall, larger clinical trials comparing the added benefit of DC vaccination to that of the SOC therapy are necessary. Additionally, comparing different DC vaccination strategies as well as treatment combinations accompanied by evaluation of both markers to identify patients that will respond and patients that respond to the therapy would benefit the field largely to understand how DC vaccination therapies can be further improved.

## 6. Potential Reasons for Poor Outcome of DC Vaccines in OC

Overall, the response rates to DC vaccination in different cancer types vary largely with the average reported response rates ranging from 10–15% [94]. In OC the poor response rate to DC vaccination could be attributed to several aspects addressed below.

### 6.1. Cancer Immunoediting, Antigen Loss, HLA Polymorphisms, and Defective Antigen-Presenting Machinery

The hypothesis of cancer immunoediting suggests that the immune system keeps tumor growth under control until tumor cells evade the immune system thereby producing symptomatic cancer [84]. In all DC vaccination clinical trials and pilot studies in OC, four antigen categories were used to load DCs. These included whole tumor lysates, TA mRNA, patient autologous tumor neoantigens, and known OC TAs. Nearly half of the studies used TA peptides including Her2/neu [106,107], hTERT [108], PADRE [108], FRα [103], mucin 1 [107], p53 [105], and WT1 [106]. This strategy of antigen loading resulted in promising effects in the early stages of treatment. The phenomenon of in vivo epitope spreading can also add to the benefits where CD8^+^ T cells specific to other TAs are generated in addition to the initial target antigen used to load the DCs [107]. However, a bottleneck in this strategy is the phenomenon of antigen loss whereby tumor cells can downregulate the expression of the target antigens in an attempt to evade immune control. Epigenetic silencing of the TAs is a prominent mechanism of antigen loss [122]. Furthermore, not all patients of a particular cancer type may express the same TAs.

Additionally, HLA polymorphisms are another bottleneck in using targeted TA-loaded DC vaccines across large populations as the HLA profiling of patients is necessary to ensure the HLA-antigen compatibility [115]. These obstacles can be overcome by using patient autologous predicted epitopes or whole tumor lysates, both of which could give a broader representation of the TAs at a personalized level [113]. However, the low TMB of OC [17] could be a pitfall. This could be overcome by using DC vaccines in combination with immunogenic cell death (ICD) inducers, including chemotherapies [32,96,123,124]. The OC tumor genome has also been demonstrated to rapidly evolve which results in rare and transient T-cell responses that ultimately fail to prevent disease progression [125].

Furthermore, defective antigen-presenting machinery (APM) can contribute to overall poor patient outcomes in OC [126,127]. APM is involved in TA processing and presentation to CD8^+^ T cells, in turn enabling T-cell recognition of tumor cells. In a study undertaken by Han et al., five components of the APM (TAP1, TAP2, tapasin, HLA class I heavy chain, and β2 microglobulin) were examined in the tumors of 150 invasive epithelial OC patients. Patients expressing all five APM components in the tumor had a median survival of 5.67 years compared to 2.58 years in patients with one-to-four intact APM components versus 1.44 years in patients completely lacking APM components. Moreover, the presence of intra- and peri-tumoral T cells correlated with the positive expression of the APM components [127]. Interestingly, in a study involving 157 OC patients, intact HLA class I (consisting of intact heavy chain and β2 microglobulin) could be detected in only 52% of the patients [126]. In yet another study involving 12 patients, expression of HLA-A2 was assessed in primary versus metastatic (derived from the ascites) ovarian tumor cells. HLA-A2 loss was observed to progressively increase during the course of the disease, with ~70% of metastatic tumor cells of OC patients exhibiting HLA-A2 loss. In one of the patients, complete loss of HLA-A2 was observed in the metastatic tumor cells. However, HLA-A2 restricted Her-2 specific T-cell responses were present in the PBMCs indicating T cell-mediated immune selection for HLA haplotype loss [128]. Overall, since DC vaccination is based on antigen presentation via MHC to T cells, HLA loss and other malfunctioning APM components could contribute to the poor outcome of DC vaccination in OC patients. Yet, using specific antigen-loaded DCs could pose a bigger drawback resulting in specific haplotype loss compared to whole tumor lysate-loaded DC vaccines.

### 6.2. Immunosuppressive TME in OC

The immunosuppressive immune microenvironment within ovarian tumors and ascites is a key contribution to the poor outcome of DC vaccination. Advanced OC patients in clinical remission were found to not respond to the immunogenic protein CRM197 (non-toxic diphtheria toxin) [108]. This was rather unexpected as previous studies had shown that all healthy adults and advanced cancer patients elicited strong immune responses to CRM197 [129,130]. This indicates that the strong immunosuppressive microenvironment of OC renders the DCs unresponsive even to highly immunogenic agents.

Although cDC1s are attributed to the good prognosis of OC patients, they have been shown to poorly respond to TLR3 stimulation [82]. Moreover, cDC1s are also susceptible to activation-induced dysfunction in OC [82]. Fatty acid synthase (FASN) is an enzyme involved in fatty acid synthesis and is a key player in the tumorigenesis of several cancers. FASN is particularly upregulated in OC leading to lipid accumulation in the TME. Jian et al. demonstrated that tumor-associated DCs take up the exogenous lipids which impairs their ability to efficiently prime anti-tumor T cells [131]. Furthermore, mouse breast and pancreatic tumors have been shown to impair cDC1 development by downregulating *Irf8* in cDC progenitors by increasing granulocyte colony-stimulating factor (G-CSF) secretion. The high amount of G-CSF in the TME in turn results in the expansion of granulocytes which can serve as immune suppressors, overall shifting the balance towards immunosuppression [132]. Moreover, in ID8 murine ovarian tumors and ascites, DCs were found to present an immunosuppressive phenotype with high co-expression of PD-1 and PD-L2 which increased with disease progression. PD-1 on TI DCs suppressed NF-κB mediated secretion of cytokines and maintained DCs in an immature and suppressive state [133]. AXL expression of ID8 tumor cells was also shown to inhibit the accumulation and activation of CD103^+^ cDCs in the tumor [134]. Interestingly, tumor-derived extracellular vesicles from OC were shown to carry Arginase-1 to DCs in tdLNs, thereby inhibiting DC-mediated T-cell proliferation [49]. In addition, DCs in OC can be rendered immunosuppressive by the TME via Satb1 overexpression leading to an enhanced secretion of pro-tumoral Galectin-1 and IL-6 by the DCs [135].

Next to the findings in OC, in cancer in general, mature DCs are known to migrate from peripheral tissues into the lymph nodes where they prime T-cell responses [136]. For instance, intra-tumoral cDC1 vaccination of 1956 mOVA fibrosarcoma in *Irf8 +* 32^−/^^−^ mice that completely lack cDC1s led to efficient antigen uptake by cDC1s, which were then able to migrate to the LNs and directly elicit CD8^+^ T-cell responses. The resulting T-cell responses were strong enough to eliminate tumors at the site of vaccination as well as those growing simultaneously on the opposite flank [137]. In addition, the presence of intratumoral CD103^+^ DCs in genetically induced melanoma was reported to be necessary for recruiting effector T cells into the tumor [138]. However, the TME can negatively influence the recruitment of DCs into the tumor. In human and mouse melanoma, β-catenin expressing tumors reduced the expression of CCL4 and in turn, caused lower cDC1 recruitment into the tumor. Next to DC recruitment, IT DC activation is also impaired in the TME. For instance, ICD-induced immune activation by chemotherapy is dependent on the alarmin high mobility group protein (HMGB1), which in turn mediates DC sensing of nucleic acid from dead cancer cells. However, in murine MC38 colorectal tumors and 3LL Lewis lung tumors, high expression of TIM3 prevented this phenomenon by sequestering HMGB1 [139]. Interestingly, the expression of CD47 on MC38 tumor cells was shown to inhibit the detection of cancer cell mitochondrial DNA by signal transducer and activator of transcription-α (SIRPα) on cDC2s [140]. Hence, TME-induced restricted recruitment of DCs into the tumor as well as the suppression of DC activation could also hold true in OC. In addition to local IT DC suppression, immunosuppressive DCs can also migrate to the LNs wherein they can lead to the accumulation of Tregs and establish a metastatic niche for lymph node metastasis [141]. Therefore, not only within the tumor, but also in the tdLNs, DC function can be inhibited by the presence of a tumor.

Besides the dysfunction of DCs themselves, other immune cells in the TME can also play a role in poor DC vaccination outcomes. Tregs have been demonstrated to be key indicators of poor prognosis in OC [142]. Even if DCs are functionally potent in priming CD8^+^ T cells, a high abundance of Tregs in the TME could serve as a barrier. Additionally, tumor-associated macrophages (TAMs) and tumor cells in OC secrete high levels of CCL22 which is a Treg recruiting chemokine. Intra-tumoral Tregs in OC are highly activated and upregulate the expression of 4-1BB, OX40, and ICOS [143]. Furthermore, IL-10-producing B cells (also known as regulatory B cells or Bregs) can be found in higher abundance in the ascites compared to peripheral blood of OC patients. Breg abundance positively correlates with that of Tregs and is inversely correlated with CD8^+^ T-cell abundance. Moreover, *ex vivo* assays have shown that Bregs can suppress IFNγ secretion by T cells (even after anti-CD28 stimulation). Hence, Bregs could also serve as an indirect hurdle in the optimal stimulation of T cells by DCs [144].

Additionally, immunosuppressive M2-like TAMs are the predominant macrophage phenotype in OC. M2-like TAMs aid in OC tumor progression, metastasis, and therapy resistance. They actively attract Tregs to the TME by secreting IL-10, IL-6, TGFβ, CCL18, and CCL22. Moreover, they impair the cytotoxic functions of NK and CD8^+^ T cells in addition to promoting T-cell anergy. Additionally, they block DC maturation in an IL-10-dependent manner [145]. Interestingly, cancer stem cells in OC can increase CCL2, COX2, and prostaglandin E2 levels which can enhance macrophage polarization towards the M2-like phenotype [143].

Other prominent pro-tumoral cells in OC include immature monocytes and granulocytes (also referred to as MDSCs) that are known to be key contributors to the immunosuppressive TME of OC. They can hamper T-cell differentiation by secreting arginase-1 and can inhibit T cell and NK-cell activities by producing reactive oxygen species (ROS) and nitric oxide [146]. In OC, a high abundance of immature monocytes and granulocytes in the tumor, ascites, or peripheral blood correlates with poor prognosis [143,147]. Similarly, a high immature monocyte and granulocyte to DC ratio correlates with reduced OS [148].

Hence, the wide and intricate immunosuppressive web in OC can be a major underlying cause for the minimal outcomes of DC vaccines.

### 6.3. Meagre Outcomes of moDC Vaccination

Apart from the immunosuppressive TME, the DC subtype used for vaccination plays a critical role in the clinical outcome. All of the DC vaccination clinical trials/pilot studies undertaken to date in OC use moDCs and have resulted in clinical benefits in only a small subset of patients [91]. This could be due to the high plasticity of moDCs and their ability to acquire an immunosuppressive phenotype *in vivo*. Several tumor- and immune-cell-derived factors (such as prostaglandin E2, ROS, and IL-6) can impair moDC development, survival, and function, often leading to an immunosuppressive phenotype. In breast cancer, colon cancer, and leukemia, moDCs serve as potent inducers of Tregs, while poorly stimulating allogeneic T cells [49].

MoDCs in mouse LLC lung cancer tumors were shown to possess superior ability in antigen uptake *in vivo* and *in vitro* compared to cDCs. However, they were also shown to be poor inducers of naïve antigen-specific CD8^+^ and CD4^+^ T-cell proliferation. In fact, MoDCs suppressed the proliferation of polyclonally or antigen-stimulated T cells in a nitric oxide-dependent manner [53]. MoDCs from the peripheral blood of melanoma patients can also hamper T-cell immunity due to their elevated expression of PD-L1 [149]. Additionally, both *in vitro* differentiated and natural moDCs do not show CCR7 upregulation in tumors, indicating their inability to migrate to the tdLN [53,150]. Toniolo et al. have shown that moDCs isolated from the blood of chronic lymphocytic leukemia patients showed poor activation status (CD40, CD80, CD83, and CD86), even after stimulation with LPS, compared to healthy donors. Moreover, the moDCs suppressed CD4^+^ and CD8^+^ T-cell proliferation and induced the expansion of Tregs [151]. Another reason for the poor effectiveness of moDC vaccines could be their dependence on host endogenous DCs for T-cell stimulation. This is because moDCs have been shown to serve as a vehicle that transfers the antigens to host cDCs, which are responsible for carrying out direct CD4^+^ and CD8^+^ T-cell stimulation [137].

With our current understanding of the high plasticity of moDCs, using them as DC vaccination agents may not be the optimal option. Even if they can be stimulated to acquire an anti-tumoral phenotype, they are not likely to maintain the same phenotype within the TME [43]. In fact, long periods of *ex vivo* differentiation could be one of the reasons for their exhaustion and poor migratory capacity upon re-infusion *in vivo* [150]. With that said, so far clinical trials have largely focused on moDCs, particularly due to the ease in retrieving monocytes in large numbers via leukapheresis from peripheral blood and further *in vitro* differentiation and expansion of moDCs. Hence, looking beyond moDCs into other DC subsets is necessary for improving the therapeutic benefits of DC vaccines.

## 7. Future Perspectives

While the current approaches to DC vaccination are steadily progressing, a new outlook toward different steps in the vaccine manufacturing procedure is necessary. Using natural cDCs instead of *in vitro* differentiated moDCs could prove therapeutically beneficial especially since the brief *ex vivo* culturing periods of the former could enable them to retain their functional capacities and avoid exhaustion [150]. Wculek et al. performed B16-OVA melanoma whole tumor cell lysate-loaded splenic cDC1 vaccination in mice in both prophylactic and therapeutic settings. In both settings, the tumor progression was significantly reduced and survival was improved [152]. Similarly, Ferris et al. used *in vitro* generated cDC1s (cultured in FMS-like tyrosine kinase 3 ligand (Flt3L) containing medium) from bone marrow to vaccinate 1956 fibrosarcoma mOVA bearing mice. To this end, the cDC1s were injected intratumorally in *Irf8* + *32^−/−^* mice (that lack cDC1s), which resulted in tumor regression. The cDC1 vaccine was also tested in 1956 fibrosarcoma which lacked OVA expression. Yet tumors were still rejected, hinting toward the potential of cDC1s to induce anti-tumor immunity even in the absence of a strong TA. Moreover, the cDC1s used in this study were not loaded with antigen but were directly injected into the tumor wherein they exhibited superior ability to take up antigen, migrate to the tdLN and elicit an immune response. Interestingly, the immune response generated was systemic, resulting in abscopal effects [137]. As cDC1 abundance in OC is inversely correlated to the CA125 levels, cDC1 vaccination in OC may be a potential option [82].

Conversely, cDC2s vaccination in 1956 fibrosarcoma did not result in potent immune responses [137]. However, the cDC2 heterogeneity in cancer is profound, both at the phenotypic and functional levels [37,39,79]. Hence, an in-depth characterization of the different cDC2 subsets in OC is indispensable. From here, the most functionally appropriate cDC2 subset/s can then be used as a vaccine candidate.

An apparent bottleneck in using natural DCs is their scarcity in circulation. This pitfall can be alleviated by boosting the DC numbers using recombinant Flt3L administration in patients. Flt3L is a growth factor that expands DC populations. Several pre-clinical and clinical trials have demonstrated that Flt3L can provide therapeutic benefits against cancer by improving anti-tumor immunity, particularly when used in combination with other therapies [48,153]. Pre-surgery administration of Flt3L into metastatic colon cancer patients led to an expansion of blood and perilesional DCs [153]. Moreover, subcutaneous administration of Flt3L in carcinoembryonic antigen cancer patients led to a 5–10 fold increase in circulating DCs [153,154]. Flt3L administration has also been shown to aid in tumor rejection in a cDC1-dependent manner when combined with immunotherapy [38,48]. Currently, 13 clinical trials using recombinant Flt3L (in combination with other therapies) in different cancer types have been registered on ClinicalTrials.gov (accessed on 15 August 2022).

Besides the meager outcomes of DC vaccination in OC, the extremely costly and complex manufacturing process involving long periods of *in vitro* culturing steps hinders the use of the DC vaccine across the wider public. Hence, autologous tumor-associated DCs derived from tumor tissue [53] or ascites [155] could also serve as potent candidates for vaccination. Mature tumor-associated cDC1s and cDC2s from LLC-OVA lung and B16-OVA melanoma tumors that were neither *in vitro* stimulated nor antigen-loaded were used to vaccinate mice prophylactically. While cDC2 vaccination resulted in slower tumor growth in LLC-OVA-bearing mice, cDC1 vaccination was more beneficial in B16-OVA-bearing mice. This study puts forth the idea that cDCs in the TME have already taken up the TAs *in vivo* and do not require additional *ex vivo* activation and can therefore serve as potential candidates for DC vaccination [53]. In addition, Adams et al. have demonstrated that upon *ex vivo* maturation with TLR-agonists and IL-10R blocking antibody, ascites-associated myeloid APCs (CD11b^+^F4/80^+^CD11c^+^) from the ID8 mouse model could efficiently induce OC tumor clearance in a T cell-dependent manner when used as a prophylactic vaccine. Moreover, CD14^+^ monocytes from peritoneal ascites of OC patients actively presented autologous TAs. In addition, human ascites-derived CD14^+^ monocytes upon stimulation with LPS and IL-10R and co-cultured with ascites-associated lymphocytes efficiently led to strong Granzyme, A, Granzyme B, and IFNγ secretions [155]. Even though Adams et al. use monocytes as a vaccination agent rather than DCs, overall these studies suggest that retrieval of the DCs from the tumor microenvironment (tumor tissue or ascites) could avoid the need for *ex vivo* antigen loading during vaccine manufacturing [53,137,155] thereby considerably reducing the cost of DC vaccination.

Tumor load and response rate of patients to therapy at the time of inclusion in the clinical trial/pilot study also play a key role in determining the success of the DC vaccine. Interestingly, patients with progressive disease at the time of enrollment into the study responded poorly/did not respond to DC vaccination, while those with stable disease showed enhanced clinical outcomes [101]. Furthermore, DCs in OC have been shown to turn immunosuppressive as the disease progresses [84]. Tumor-load-mediated immune suppression is known to be one of the root causes of DC vaccine failure in cancer. Hence, using DC vaccines as an adjuvant therapy may be an interesting option. In 71% of melanoma patients, antigen-specific T cells were detected in response to adjuvant DC vaccination versus 23% following vaccination in the metastatic setting [94]. Since immunosuppression is high in OC, administering DC vaccines only at the moment where this is decreased by the state of the art might be beneficial [156].

Besides DC vaccination strategies employing patient autologous DC retrieval and subsequent re-administration, *in vivo* DC targeting approaches can also be further investigated in OC. Direct *in vivo* antigen delivery specifically to DCs is one such approach [157,158,159]. As DCs can to some extent be identified uniquely using cell surface markers, monoclonal antibodies targeting these receptors could be used to shuttle antigens specifically to the DCs [157,158,159]. Several pre-clinical studies have been undertaken to target DCs via specific markers including DEC-205, CD11c, Clec9A, LOX1, mannose receptor (MR), CD36, and CD317 [157]. Five clinical trials for *in vivo* targeting of DCs in OC have been registered on ClinicalTrials.gov (accessed on 15 August 2022). NCT01522820 and NCT00948961 used CDX-1401 (in combination with other drugs) which is a human monoclonal IgG1 linked to the tumor antigen NY-ESO-1 and targeting DCs via DEC-205. However, while the treatment was well tolerated, no clear clinical benefit was seen in OC patients [160]. In NCT00648102 and NCT00709462, CDX-1307 was used which contains a human monoclonal antibody targeting mannose receptors on DC linked to the human chorionic gonadotropin-β chain. While the therapy was able to induce humoral and T-cell immunity, only nine out of 87 cancer patients had stable disease for 2.3 to 18.2 months; however, the cancer type of these patients is not known [161]. Finally, NCT02387125 uses LV305, a DC-SIGN mediated DC-targeting engineered lentiviral vector consisting of nucleic acids encoding NY-ESO-1. LV305 was used in combination with a DC-activating agent called G305 which consists of NY-ESO-1 recombinant protein and TLR4 targeting glucopyranosyl lipid adjuvant-stable emulsion. However, this study was terminated as it did not meet the efficacy objective. Overall, *in vivo* DC targeting in OC is still in an early stage of development and further studies need to be performed to understand their applicability in OC.

Next to *in vivo* DC targeting, there is room to explore other strategies for improving *in vivo* DC activation, trafficking, and survival in OC patients. For the efficient presentation of antigens by DCs, besides antigen uptake, DC maturation is crucial, which often occurs through environmental signals called pathogen-associated molecular patterns (PAMPs) and damage-associated molecular patterns (DAMPs). Importantly, in specific TA targeting strategies, mere antigen uptake by DCs in the absence of PAMPs and DAMPs can render them tolerogenic against the said TA [159,162]. Commonly used *in vivo* DC activating agents in cancer are agonists for TLR3, TLR7, TLR8, and TLR9 which serve as adjuvants to enhance DC-mediated T-cell stimulation [159]. CD40 is a co-stimulatory receptor on the surface of DCs and other APCs, which upon engaging with its ligand can boost MHC molecule production, pro-inflammatory cytokine secretion, and T-cell stimulation [159]. Hence, using CD40 ligand (CD40L) as an *in vivo* DC activation method appears promising. Indeed, in mouse and human pancreatic ductal adenocarcinoma, CD40 agonist administration was shown to enhance T-cell infiltration into cold tumors, mediate T-cell clonal expansion, and reduce IT Treg infiltration in turn eliciting a durable anti-tumor immune response [163]. Since ovarian tumors have been shown to sometimes be immune deserts, largely lacking T cells [164], CD40 agonistic therapy could be beneficial. Additionally, GM-CSF is also employed as a DC maturation and recruiting agent in anti-cancer therapies. GM-CSF pre-treatment in bladder and prostate cancer has been shown to enhance the number of circulating DCs as well as CD4^+^ and CD8^+^ T cells, increase the recruitment of CD8^+^ T cells into the tumor and reduce the accumulation of IT myeloid suppressor cells [165,166]. Hence, in addition to DC vaccination, DC targeted therapies and their different combinations hold great potential as therapeutic advancements against OC. That said, it is important to consider the immune biology of OC before devising such therapies. For instance, overstimulation of cDC1s by external agents could render them dysfunctional, as has been previously demonstrated in OC [82]. Additionally, pDCs from breast tumors in the presence of GM-CSF were shown to turn tolerogenic and advance tumor progression [78].

Overall, current advances in DC vaccination against cancer are still sub-optimal with a meager objective response rate of only 10–15%. There is still a lack of Phase III clinical trials of DC vaccination, with a majority of the clinical efficacy data currently coming from Phase I/II clinical trials that are based on short-term criteria. Adding to that, DC vaccination strategies currently used are not homogenous and vary at multiple levels, ranging from the DC subtype used, vaccine manufacturing process, antigen type, route of administration, and combination therapies. This in turn complicates making comparisons across studies. Moreover, there is a lack of reliable predictive biomarkers to determine the therapeutic efficacy of DC vaccines, which could be used to evaluate their real therapeutic efficacy [32].

## 8. Conclusions

In conclusion, DC vaccination in OC holds immense potential to improve patient outcomes. Traditional tumor-targeted cytotoxic therapies provide only transient responses, which are lost when the therapy is discontinued. DC vaccination on the other hand could provide enduring responses by mobilizing the immune system to target the tumor, which could also enhance anti-tumor memory. This would be a critical aspect in preventing metastasis and relapse [167]. However, further research is still necessary to streamline the understanding of DC heterogeneity in OC which could be harnessed to develop therapies resulting in durable objective responses across wider OC patient cohorts, thereby resulting in long-term patient survival.

## Figures and Tables

**Figure 1 cancers-14-04037-f001:**
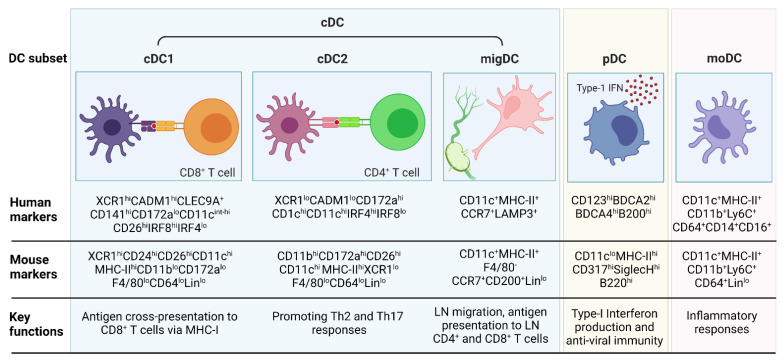
Overview of the different DC subsets, their corresponding mouse and human markers, and key functions. Lin = CD3^+^B220^+^CD19^+^NK1.1^+^; LN, lymph node. Figure created with Biorender.com.

**Figure 2 cancers-14-04037-f002:**
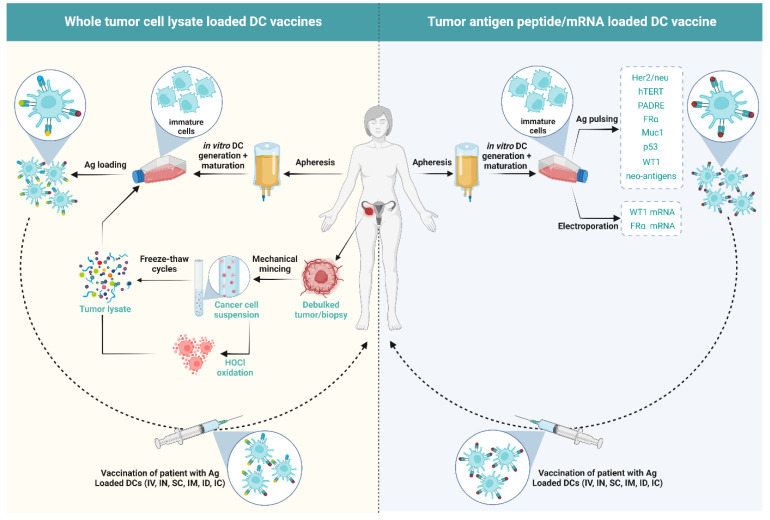
Schematic overview of dendritic cell vaccination strategies used in ovarian cancer. Ag, antigen; HOCl, hypochlorous acid; IV, intravenous; IN, intranodal; SC, subcutaneous; ID, intradermal; IC, intracutaneous. Figure created with Biorender.com.

**Table 1 cancers-14-04037-t001:** Overview of the clinical trials on DC vaccination against OC registered on ClinicalTrials.gov (accessed on 15 August 2022).

Status	NCT Number	Phase	Treatment|Combinations
Active, not recruiting	NCT00799110	Phase 2	DC/tumor fusion vaccine|Imiquimod|GM-CSF
Active, not recruiting	NCT02033616	Phase 2	DC vaccine|PBMCs
Active, not recruiting	NCT02111941	Early Phase 1	DC vaccine
Not yet recruiting	NCT05270720	Phase 1	DC vaccine
Not yet recruiting	NCT03735589	Phase 1|Phase 2	DC vaccine|Autologous NK Cell-like CTLs
Recruiting	NCT00703105	Phase 2	DC vaccine
Recruiting	NCT04614051	Phase 1	DC vaccine
Recruiting	NCT04739527	Phase 1	DC vaccine
Recruiting	NCT04834544	Phase 2	DC vaccine
Completed	NCT01617629	Phase 2	DC vaccine
Completed	NCT01068509	Phase 2	DC vaccine
Completed	NCT01132014	Early Phase 1	DC vaccine
Completed	NCT00478452	Phase 1	DC vaccine|DC vaccine with Cyclophosphamide
Completed	NCT00683241	Phase 1	DC vaccine
Completed	NCT00005956	-	HER-2/neu intracellular domain protein|DC vaccine
Completed	NCT03657966	Phase 2	DC vaccine|Standard-of-care chemotherapy
Completed	NCT00004604	Phase 1	DC vaccine
Completed	NCT00027534	Phase 1	DC vaccine
Completed	NCT00019084	Phase 2	Aldesleukin|DC vaccine|Ras peptide cancer vaccine|Sargramostim|Autologous lymphocytes|Therapeutic tumor-infiltrating lymphocytes
Completed	NCT02179515	Phase 1	DC vaccine
Completed	NCT01132014	Early phase 1	DC vaccine
Completed	NCT02107950	Phase 2	DC vaccine in parallel with chemotherapy|Standard-of-care
Completed	NCT02107937	Phase 2	DC vaccine with Standard-of-care|DC vaccine after chemotherapy|Standard-of-care
Unknown status	NCT01456065	Phase 1	DC vaccine

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
