# Peer review of "Dendritic Cell Vaccines: A Promising Approach in the Fight against Ovarian Cancer"

_cancers, 2022, doi:10.3390/cancers14164037_

Round 1

Reviewer 1 Report

The manuscript by Caro et al reviews the role of dendritic cell vaccines in therapy against ovarian cancer. While the manuscript is generally well written and provides an extensive coverage of the topic, some sections (i.e. DC subset description) could be condensed to improve the focus of the review.

Specific suggestions/comments to improve the manuscript include:

( (1)    Line 59 – “has fired up” should be replaced with a more scientific description.

  (2)    Lines 80-91 – given that ACT has been mentioned as a treatment, the authors should also briefly cover CAR T cell therapy for ovarian cancer.

  (3)     Line 97 – did this study investigate how DC vaccination caused the Treg cell number to reduce? If so, the mechanism should be briefly mentioned.

  (4)    The whole of section 3 could be condensed significantly.  The subtypes of DCs (plus markers) could be covered in a Table and their important features only briefly mentioned in the text.  This will help maintain a focus on DCs within ovarian cancer.

  (5)    Line 351-352 – the comparison here is Arm 3 with Arm 1 (not Arm 2 as written)

  (6)    Section 6.1 – is HLA loss also seen in ovarian cancer.  If the tumour loses HLA, can the DCs in the tumour microenvironment still activate a local immune response ? 

  (7)    Section 6.2 – where do the DCs have their key effects on T cells, is it in the draining lymph node or in the tumour microenvironment?  Is it immunosuppressed DCs heading to the lymph node that is the problem or DCs residing in the suppressed tumour environment affecting T cell reactivation or both. There is an opportunity in this section to dissect different roles for DCs based on location.

   (8) Section 7 – could include ideas on improving DC survival and trafficking in cancer patients.  In addition, while the review is focused on DC harvest and return to the patient, what about constructs which target tumour antigen to the DCs in vivo (ie. DC targeting antibodies with attached tumour antigen) or selectively activate the DCs in vivo.

Author Response

(1) Line 59 – “has fired up” should be replaced with a more scientific description.

Response : We agree with the reviewer and adapted the sentence to “The interest in immunotherapy for OC treatment has been boosted, particularly since it was shown that intra-tumoral (IT) T cells, especially CD8+ T cells, in treatment naïve OC patients were associated with improved clinical outcomes and survival”.

(2) Lines 80-91 – given that ACT has been mentioned as a treatment, the authors should also briefly cover CAR T cell therapy for ovarian cancer.

Response : We introduced a section covering the currently used CAR T-cell therapies in OC in lines 97-151.

(3) Line 97 – did this study investigate how DC vaccination caused the Treg cell number to reduce? If so, the mechanism should be briefly mentioned.

 Response : The study by Correll et al performed was a correlative study based on melanoma progression, Treg frequency in the blood and overall T-cell responses. Unfortunately, Correll et al were not able to perform functional assays to investigate the mechanisms, because the frequency of Tregs in the blood and the volume of patient blood samples they received were too low.

Hence, we did not adapt the paragraph about this study.

(4) The whole of section 3 could be condensed significantly.  The subtypes of DCs (plus markers) could be covered in a Table and their important features only briefly mentioned in the text.  This will help maintain a focus on DCs within ovarian cancer.

 Response : We agree with the reviewer. We strongly condensed the text and provided an overview of the panel of markers used to identify the different DC subsets in mice and humans as well as their key functions in a new figure (Figure 1)

(5) Line 351-352 – the comparison here is Arm 3 with Arm 1 (not Arm 2 as written)

 Response : We thanks the reviewer for this comment and corrected our mistake.

(6) Section 6.1 – is HLA loss also seen in ovarian cancer.  If the tumour loses HLA, can the DCs in the tumour microenvironment still activate a local immune response ? 

Response : We added a paragraph discussing HLA-loss in OC. We also clarified in the paper that since DC vaccination is based on antigen presentation via MHC to T cells, HLA loss and other malfunctioning components of the antigen-presenting machinery could contribute to the poor outcome of DC vaccination in OC patients. Consequently, using specific antigen-loaded DCs could pose a bigger drawback resulting in specific haplotype loss compared to whole tumor lysate-loaded DC vaccines (lines 834-855).

(7) Section 6.2 – where do the DCs have their key effects on T cells, is it in the draining lymph node or in the tumour microenvironment?  Is it immunosuppressed DCs heading to the lymph node that is the problem or DCs residing in the suppressed tumour environment affecting T cell reactivation or both. There is an opportunity in this section to dissect different roles for DCs based on location.

Response : We included a paragraph describing how the TME of OC can maintain DCs in a suppressed state (lines 913-918). Regarding the function of DCs based on location, to our knowledge, not much has been reported in OC. Therefore, to reply to re reviewer’s comment, we also included findings from other cancer types (lines 923-946).

(8) Section 7 – could include ideas on improving DC survival and trafficking in cancer patients.  In addition, while the review is focused on DC harvest and return to the patient, what about constructs which target tumour antigen to the DCs in vivo (ie. DC targeting antibodies with attached tumour antigen) or selectively activate the DCs in vivo.

Response : We included a section discussing in vivo DC targeting approaches and strategies for improving in vivo DC activation, trafficking, and survival in OC patients (Lines 1122-1180).

Reviewer 2 Report

Author Caro et al. through the review article, “Dendritic cell vaccines: a promising approach in the fight against ovarian cancer” have emphasized on the potential of Dendritic cell (DC) vaccines. They have presented a broad overview of immunotherapies in ovarian cancer and have discussed different subsets of DC and their involvement in OC. The review summarizes clinical trials undergoing or in the past and have stressed on the possible development of DC vaccine-based therapeutics. They have tried to provide a balance opinion by discussing the limitations of DC based therapeutics and the improvement possibilities in the future.  

The review is interesting to read and clinically relevant. It can be considered for publication. However, one minor comment needs to be addressed before publication.

Minor comment:

1.     Line 330, the date of database access for vaccination clinical trials in OC registered on ClinicalTrials.gov is not provided for table 1. It should be included in the text.

Author Response

Minor comment:

  1. Line 330, the date of database access for vaccination clinical trials in OC registered on ClinicalTrials.gov is not provided for table 1. It should be included in the text.

Response : We agree with the reviewer and added the date of database access in all lines mentioning data from ClinicalTrials.gov.